# The Tunable Rhenium Effect on the Creep Properties of a Nickel-Based Superalloy

**DOI:** 10.3390/ma17010191

**Published:** 2023-12-29

**Authors:** Xiao-Zhi Tang, Ya-Fang Guo

**Affiliations:** Department of Mechanics, School of Physical Science and Engineering, Beijing Jiaotong University, Beijing 100044, China; yfguo@bjtu.edu.cn

**Keywords:** intermetallics, creep, alloying elements, dislocations, molecular dynamics

## Abstract

Atomistic simulations on the creep of a nickel-based single-crystal superalloy are performed for examining whether the so-called rhenium effect can be tuned by changing the spatial distribution of rhenium in the nickel matrix phase. Results show that Rhenium dopants at {100} phase interfaces facilitate mobile partial dislocations, which intensify the creep, leading to a larger creep strain than that of a pure Ni/Ni3Al system containing no alloying dopants. If all the Re dopants in the matrix phase are far away from phase interfaces, a conventional retarding effect of Re can be observed. The current study implies a tunable Re effect on creep via dislocation triggering at the phase interfaces.

## 1. Introduction

Nickel-based single-crystal superalloys are one of the major important high-temperature materials that are extensively used for turbine blades in aircraft engines. They are generally composed of a γ-γ′ two-phase microstructure, which endows the superalloy with superior mechanical properties at elevated temperatures. The γ phase is a disordered face-centered cubic Ni matrix, and the γ′ phase is an ordered L1_2_ Ni_3_Al precipitate. The γ′ phases are in cubicle shapes and embedded in γ phase, taking a high volume fraction around 60–70% [1,2]. At the γ/γ′ phase interfaces, interfacial dislocation networks are generated by misfit stress, which arises from the different lattice parameters between Ni and Ni_3_Al [3,4]. These interfacial dislocation networks are located on three possible phase interfaces: {100}, {110}, and {111} [4,5]. Among them, the {100} interface is prevalent during steady-state creep at high temperatures in this two-phase microstructure [6]. 

Modern commercial Ni-based superalloys usually contain more than ten alloying elements from their first generation. Among them, rhenium characterizes different generations of Ni-based superalloy depending on its fraction [7,8]. Re is mostly distributed in the γ phase and increases the flow stress by solid solution strengthening, as it has a larger atomic radius than Ni does [9]. It is also found to segregate at the γ/γ′ phase interfaces during the creep process [10]. In particular, it is enriched in the tensile stress regions of interfacial dislocation cores [3,9]. Since the coarsening of the γ′ phase during the creep process of Ni-based superalloy is accompanied by the movements of interfacial dislocation networks, Re distribution in the γ phase is supposed to be inherently related to its effects on the mechanical responses [11]. 

In this article, the effects of Re distribution in the γ phase on the creep deformation of Ni-based single-crystal superalloys is investigated. For changing Re distribution easily and investigating the creep behavior efficiently, a molecular dynamics (MD) simulation is adopted. 

Until now, there are only a few of MD studies which investigated the creep behavior of Ni-based superalloy [12,13,14,15]. To authors’ best knowledge, none of them considered the variation of Re distribution in the γ phase. In 2016, Yu et al. adopted MD to simulate the creep behavior of Ni-based superalloy using a two-dimensional (2D) system without any alloying dopants [12]. In 2021, Khoei et al. did the same thing but at much wider ranges of temperature and stress [13]. In 2022, the same authors of Ref. [13] repeated similar simulations using a three-dimensional (3D) system specifically focusing on the interfacial dislocation network evolution in a 2.5 GPa stress state. Their model in that work did not have a precise volume fraction, but had Re additions in the Ni matrix [14]. In the same year, Wu et al. also investigated the Re effect on creep properties [15]. However, they used a different 3D system which has a cubic-shaped Ni_3_Al precipitate surrounded by the Ni matrix, so their model has the correct volume fraction. Wu’s simulation had a longer creep time than others’, and the time was up to nanoseconds. A comparison between these studies and ours is listed in Table 1. The present study has three advantages compared with previous ones: the correct volume fraction, alloy additions (Re dopants) in the model, and consideration of dopant distribution. The results of the present study are expected to reveal whether the Re effect on the creep of Ni-based superalloys is tunable, by changing the distribution of Re in the γ phase.

## 2. Materials and Methods

To mimic a real single-crystal Ni-based superalloy in simulations, a 3D model including both the γ phase and γ′ phase is established, as shown in Figure 1a. The whole model has dimensions of approximately 30 nm × 30 nm × 30 nm, containing 2,470,000 atoms, and the periodic boundary condition is applied along all three Cartesian coordinates. The γ′ phase has a dimension of approximately 27 nm × 27 nm × 27 nm, resulting in a volume fraction of about 69%, which is comparable to that in experiments (~70%) [17]. The γ/γ′ phase interfaces are {100}, and the x, y and z axes are [100], [010], and [001], separately.

The inevitable lattice misfit (δ) at the γ/γ′ phase interface due to the different lattice parameters between Ni and Ni_3_Al has its largest value, according to its definition of $\delta =2(\alpha_{\gamma^{\prime }}-\alpha_{\gamma })/\alpha_{\gamma^{\prime }}+\alpha_{\gamma },$ under the conditions of a one-to-one unit cell. Such a misfit is optimized by different lattice numbers of two phases, following a relationship of $n\alpha_{\gamma \prime }=(n+1)\alpha_{\gamma }$. According to experiments and *ab-initio* calculations, we adopted $\alpha_{\gamma \prime }=3.567$ Å and $\alpha_{\gamma }=3.52$ Å, so the above relationship yields n≈75. The residual lattice misfit along with the different lattice numbers of the two phases results in a 3D interfacial 1/2<110> perfect dislocation network on the {100} interfaces. This interfacial dislocation network is presented in Figure 1b. This interfacial dislocation network pattern is the same as in previous similar MD research [15,19], and consistent with scanning electron microscope observations [4,20].

To consider Re distribution’s effects, Re alloying dopants are randomly added into the γ phase in two ways: the first one is leaving a 1 nm Re-dopant-free zone adjacent to phase interface, as Figure 1d shows; and the second one is replacing Ni atoms with Re atoms in the whole γ phase, so that interfacial Re dopants exist, as Figure 1e shows. These two ways enable us to investigate whether the interaction between Re dopants and the γ/γ′ phase interface (especially those with interfacial dislocations) affects the mechanical response during creep. In the following sections, we name simulations using the model constructed with the first method, as in Case 2 (Figure 1d), and the model constructed with the second method, as in Case 3 (Figure 1e). Case 1 is a model without Re dopants, which can be represented by the model in Figure 1b.

The widely used EAM potential for the Ni-Al-Re system [7] is adopted in the present study. This EAM potential has been benchmarked by experimental and first-principles data, and adopted to study a series of properties of Ni-based superalloys, including the alloying dopants segregations, lattice trapping, interface crack propagation, etc. [3,14,19,21,22,23]. It also maintains the stability of our 3D model in MD simulations. For creep simulations by MD, creep stress is defined as the pre-set stresses along three Cartesian coordinates, which are controlled by the barostat in the ensemble (similar to a Nose–Hoover style one in an isothermal–isobaric ensemble), and creep time is defined as the period during which the measured system stresses are set as constants. The common strategy for creep testing in MD simulations is to apply mechanical loading at higher temperatures and stresses than those in laboratories, so that the creep deformation can be observed at MD timescale [13,15]. The temperature adopted in the present simulations is 1500 K, which is within the applicable range of the used atomic potential and close to the service temperature of Ni-based superalloys in turbines. The stress adopted in the present simulations is 2 GPa, which is below the yielding stress of around 3 GPa at 10^8^ s^−1^ strain rate in MD simulations of uniaxial tension (performed in the present study, but not shown for concision). As listed in Table 1, most previous studies had a simulated creep time less than 1 ns, so for describing the creep responses better, the creep time simulated in present study was set at 3 ns, which is less than the longest simulated creep time of 20 ns recently used in Ref. [15], but saves lots of computation costs. The timestep adopted is 2 fs. The temperature and stress is controlled by an isothermal–isobaric ensemble via the Nose–Hoover algorithm. Creep stress is applied uniaxially, and normal stresses along the other two directions are kept zero. Simulations are performed by LAMMPS [24], which is a classical molecular dynamics simulator that enables the particle-based modeling of materials, and has potential for solid-state materials (metals, semiconductors) and soft matter (biomolecules, polymers) and coarse-grained or mesoscopic systems. Both potential energy minimization and classical molecular dynamics modules are used in current simulations in the above sequence. Atomic structure visualization is conducted using OVITO basic 3.8.5 software [25].

## 3. Results and Discussion

Figure 2a shows the strain–time relationships of the creep processes at 1500 K and 2 GPa, separately, in the following three cases: Case 1 for the pure Ni/Ni_3_Al system, Case 2 for no interfacial Re dopants, and Case 3 for present interfacial Re dopants. Surprisingly, Case 3 (blue curve) does not exhibit the expected retardation effect of Re on creep. On the contrary, after approximately 400 ps, Case 3 has a larger creep strain than Case 1 (red curve, no Re dopants, pure Ni/Ni_3_Al system). Case 2 (green curve, no interfacial Re dopants) shows a retardation effect on creep strain that is smaller than that of the pure Ni/Ni_3_Al system. The simulations are repeated in the same stress state, but at a lower temperature of 1300 K, obtaining the same results (Figure 2b). 

The only difference between Case 2 and Case 3 is whether there are interfacial Re dopants. If there are, the creep is more severe than that in Case 1 (the pure Ni/Ni_3_Al system), and if there are not, creep is milder than that in Case 1. Since in previous studies, Re dopants have been found to promote 1/6<112> partial dislocations [19], it is reasonable to speculate that interfacial Re dopants lead to higher dislocation density. Therefore, in Figure 3a, we plot the dislocation density variation. It can be seen that the initial densities of Case 2 and Case 3, when creep begins, are almost the same. But in the following creep process, the dislocation density in Case 3 is remarkably larger than that in Case 2. This discrepancy demonstrates that the Re dopants at the phase interface induce more dislocation nucleation during creep, which finally leads to larger deformation.

To clearly see what happens inside the alloy (Figure 3b,c), the dislocation extraction algorithm (DXA) [26] is used to extract dislocation lines in both the γ and γ′ phases, and the CNA method is used to identify the stacking faults in the γ′ phase. It is seen that there are more dislocations (green lines in the figure) and more stacking faults in Case 3 (Re dopants in contact with the phase interface) than in Case 2 (in which the Re dopants are far away from the phase interface). The snapshots are taken at the same creep time of around 1500 ps. In Figure 3b,c, all γ-phase atoms are not shown, and only stacking-fault atoms are shown in the γ′ phase. So, it is confirmed that γ′ phase is more severely cut in Case 3 than in Case 2. For instance, there is a huge stacking fault area, as the white arrow indicates in Figure 3b. Those stacking faults are the products of dislocations which cut into the γ′ phase from γ phase, and significantly increase creep strain. 

To find out which kinds of dislocation those interfacial Re dopants promote, we separately plot the evolution of densities of different types of dislocation in the creep process in Figure 4a of Case 2 and Figure 4b of Case 3. It is found that interfacial Re dopants mainly promote 1/6<112> partial dislocations. This result is consistent with the conclusion of Ref. [19] that Re dopants promote mobile 1/6<112> partial dislocations. The structure evolution of Case 3 shown in Figure 4c reveals that interfacial Re dopants (those red atoms) around 1/2<110> interfacial dislocation (yellow lines) cores promote 1/6<112> dislocations (black lines). For instance, four white arrows identify four representative locations, where 1/6 <112> dislocations are nucleating and Re dopants close to 1/2<110> interfacial dislocations can be found. In Figure 4d, lots of 1/6<112> dislocations are witnessed at the interface in the normal to uniaxial direction, contributing to creep strain in that direction. Therefore, it is confirmed that interfacial Re dopants intensify creep deformation by promoting mobile 1/6<112> partial dislocations, leading to larger creep strain compared to the pure Ni/Ni3Al system.

Conventionally, Re dopants are believed to retard creep deformation, while in our simulations, the retarding effect is only observed under the condition that there are no interfacial Re dopants. The interfacial Re dopants eliminate the expected retardation effect on creep, and boost the creep. This result will be verified through *ab-initio* calculations and probably by creep experiments in the near future. In practice, if proper ageing treatments in metallurgical processes are adopted to slow down the diffusion of Re dopants towards the tensile regions of interfacial dislocations, creep resistance is supposed to be enhanced by avoiding unnecessary promotion of dislocations, such as that of 1/6<112> partial dislocations.

## 4. Conclusions

In this article, MD simulations are carried out to perform creep deformation on a Ni-based superalloy for examining whether Re effects can be tuned by changing Re distribution. We design two separate models, which either do or do not have Re dopants at the γ/γ′ interface, to compare creep strain with that of a pure Ni/Ni3Al system. Surprisingly, the retarding effect vanishes if there are Re dopants at the γ/γ′ interface. Furthermore, interfacial Re dopants induce larger creep deformation than that of the pure Ni/Ni3Al system by promoting mobile 1/6<112> partial dislocations. Such promotions are consistent with the observations of a previous study [19]. Therefore, it should be expected that effects of Re on the creep of Ni-based superalloys are tunable by changing dopants’ distribution in the γ phase.

## Figures and Tables

**Figure 1 materials-17-00191-f001:**
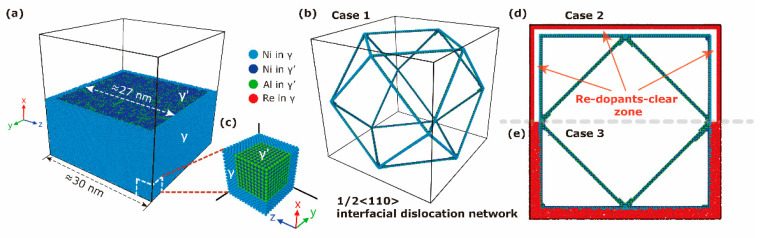
(**a**) Geometry of the atomic model of a Ni-based single-crystal superalloy (for visual clarity, the model is sliced in half). (**b**) The interfacial dislocation network identified by common neighbor analysis (CNA) [18]. The model without alloying dopants (a pure Ni/Ni_3_Al system) is referred to as Case 1 in the current simulations. (**c**) Enlarged local atomic environment of the γ/γ′ phase interface, from one corner of the simulated system as the white box indicated in (**a**). (**d**) The model with Re dopants in γ is separated from the γ/γ′ phase interface by about 1 nm (a Re-dopant-free zone), and is referred to Case 2. Here, only half of the Case 2 model is shown. (**e**) The model with Re dopants full of γ (Re dopants in contact with the γ/γ′ phase interface and the interfacial dislocations) is referred to as Case 3. Only half of the Case 3 model is shown.

**Figure 2 materials-17-00191-f002:**
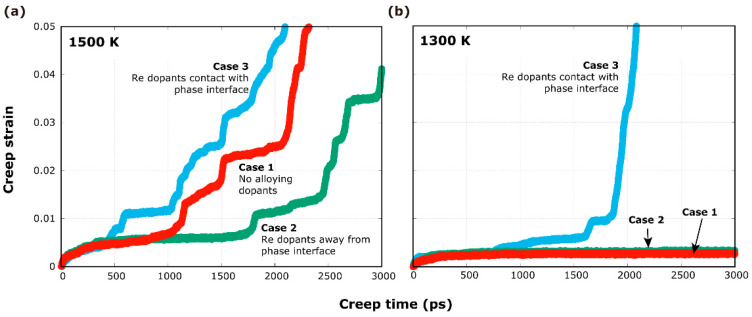
The strain–time relationships of the creep processes at 2 GPa (and separately): (**a**) 1500 K and (**b**) 1300 K of three simulated cases.

**Figure 3 materials-17-00191-f003:**
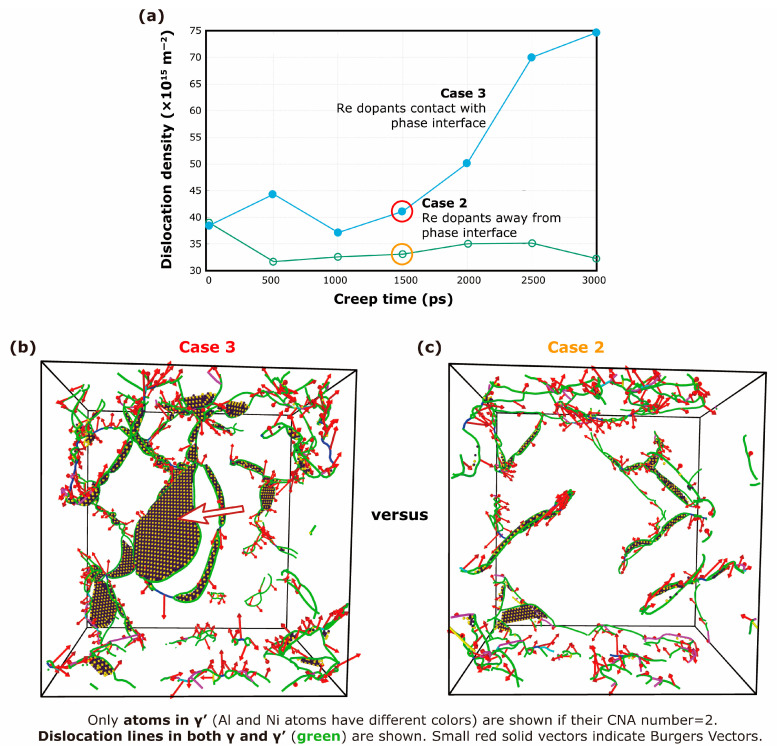
(**a**) Evolution of total dislocation densities versus time in Case 2 and Case 3. (**b**) Dislocations in the system of Case 3. The white arrow points to a stacking fault in the γ′ phase, which means the γ′ phase is severely cut by a dislocation. (**c**) Dislocations in the system of Case 2.

**Figure 4 materials-17-00191-f004:**
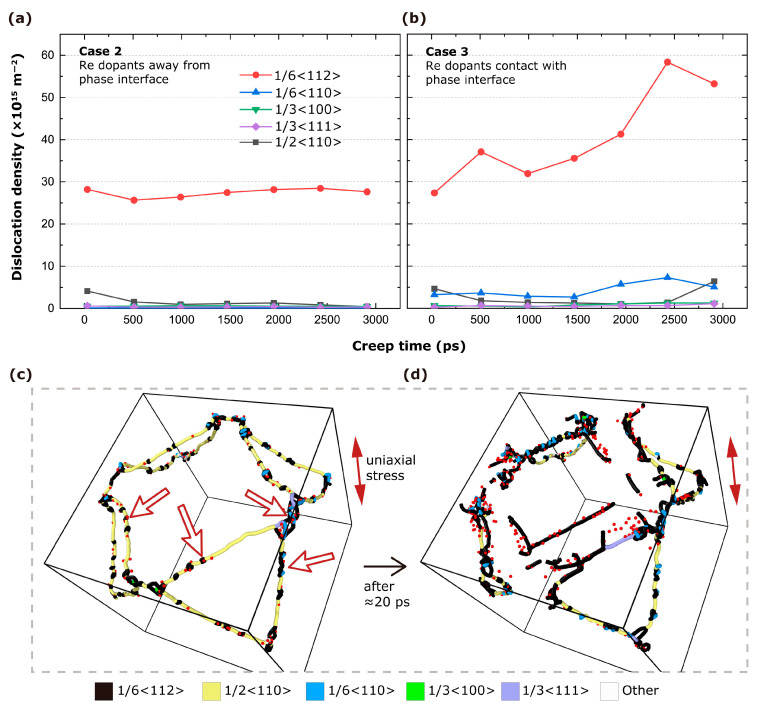
The dislocation density of different categories versus creep time, separately, in (**a**) Case 2 and (**b**) Case 3. (**c**,**d**) are the dislocation network evolution of Case 3 (time sequence: (**c**) to (**d**)), along with Re dopants (red spheres) around dislocation cores. For visual clarity, only half of the system is shown.

**Table 1 materials-17-00191-t001:** Comparison between previous MD studies on the creep properties of Ni-based single-crystal superalloys and the present one in this article.

Research	Correct Volume Fraction	Alloying Dopants	Dopant Distribution Effect	Temperature	Stress	Creep Time	Atomic Potential
Ref. [12]	No	No	No	1100 K to 1300 K	0.3 GPa to 0.5 GPa	Up to 750 ps	Not mentioned
Ref. [13]	No	No	No	1100 K to 1700 K	0.5 GPa to 5 GPa	Up to 670 ps	2009 Mishin [16]
Ref. [14]	No	Yes	No	1400 K to 1700 K	2.5 Gpa	Up to 750 ps	2012 Du [7]
Ref. [15]	Yes	Yes	No	1000 K	0.5 GPa to 2 GPa	Up to 2 × 10^4^ ps	2012 Du [7]
Present study	Yes	Yes	Yes	1300 K and 1500 K	2 GPa	Up to 3 × 10^3^ ps	2012 Du [7]

## Data Availability

Data are contained within the article.

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
