# Peer review of "The Tunable Rhenium Effect on the Creep Properties of a Nickel-Based Superalloy"

_materials, 2023, doi:10.3390/ma17010191_

Round 1
Reviewer 1 Report
Comments and Suggestions for Authors
The authors present a Molecular Dynamics - based investigation of the effects of Rhenium dopants in Nickel-based superalloys that are used, for example, for applications in turbine blades.
Specifically, the authors study the possibility to tune the Rhenium retardation effect on the creep strain by modulating the distribution of the dopant Re atoms at the interface between the \gamma and \gamma' phases.
In these preliminary study, three models are used where Re atoms are either (1) absent, (2) located at the \gamma/\gamma' interface and (3) located at a certain distance from the \gamma/\gamma' interface.
Results show the known retardation effect only in case (3), while case (2) leads instead to larger creep strain. The authors conclude that their result suggests the possibility of tuning the retardation effect by modulating the dopant distribution.
This is a well-thought study. The large-scale simulations are state-of-the-art, the analysis is convincing and overall the results support the conclusion, although further studies (especially experiments) are needed to test the conclusions reported here, which is also acknowledge by the authors.
I only have minor points.
Definition of creep stress and strain is missing.
Figure 3(b),(c). The authors should consider enlarging/restructuring the figure. I can only see tiny red arrows (only zooming in one can see that they are arrows) and green lines, while the caption indicate that "purple" and "yellow" atoms are also reported.
A discussion on how dopant distribution tuning can be achieved in practice is missing.
Comments on the Quality of English Language
Moderate english editing is needed to improve clarity of discussion.
Author Response
Reply: Thank you heartfully for your comments and approbation to our work. Thanks sincerely!
I only have minor points. Definition of creep stress and strain is missing.
Reply: Thank you for pointing out this carelessness. The definitions are replenished in the lines 109 to 112 in page three of the manuscript and colored to red. Please see the revised manuscript.
Figure 3(b),(c). The authors should consider enlarging/restructuring the figure. I can only see tiny red arrows (only zooming in one can see that they are arrows) and green lines, while the caption indicate that "purple" and "yellow" atoms are also reported.
Reply: Thank you for the comments. We have revised this figure and caption to be clearer and more concise.
A discussion on how dopant distribution tuning can be achieved in practice is missing.
Reply: Such a discussion is added into the last paragraph of section 3 in red words. Thank you again sincerely for the review.
Reviewer 2 Report
Comments and Suggestions for Authors
In this work by using a so-called LAMMPS software, the creep of Nickel-based single crystal superalloy was simulated to assess the tunability of the Rhenium effect by just changing the spatial distribution of Rhenium in the Nickel matrix phase.
In general, this manuscript is difficult to read because there are many gaps of information that are not readily available. For example, several aspects of the basic geometrical aspects of a single crystals and grouping, lattice parameters, etc. are not adequately described in comprehensive terms that could be readily understood by material science readers without accessing complementary source of information. This is also the case of LAMMPS software; the given reference [26] is a general description of many modules that are of no help if in a particular application like the present work, no details are given about what sequence or number of modules were used. I am pretty sure that your experience in implementing the Re model in LAMMPS was not straightforward, with some complications but then valuable.
In my opinion this manuscript is valuable but need a major review to fill gaps of information to facilitate a fast reader’s comprehension.
Comments on the Quality of English Language
In general, the written English is good with minor grammatical mistakes.
Author Response
Reply: Thank you sincerely for the comments and suggestions. We have revised the manuscript according to these valuable criticisms, like adding geometrical aspects of this single crystal in lines 71 to 72, and adding LAMMPS modules in lines 126 to 130. Also we made some other revisions which are wished to improve this work solid for contributing to our community. Thanks again for the time and efforts of reviewer.
Round 2
Reviewer 2 Report
Comments and Suggestions for Authors
Dear authors,
I believe that the few amendments you made are satisfactory.
Concerning the standard tool used for simulation that is LAMMPS software I suggest including in the manuscript (if you think is plausible) an attention note for readers saying something like “The LAMMPS script in Windows version xxxx written for the present investigation is available upon request”. This could be useful for PhD students and beginners.
Comments on the Quality of English LanguageThe english quality is good